

# Evaluation of some garden flowers as specialty cut flowers in Eskisehir province-Türkiye

Sibel Yiğiter[1] and İlkay Coskun[2]

[1] Department of Horticulture, Eskisehir Osmangazi University, Eskişehir, Türkiye
[2] Altinbilek Agricultural Products Licensed Warehousing Inc., Eskişehir, Türkiye

Corresponding author
Sibel Yiğiter, ssaricam@ogu.edu.tr

## ABSTRACT

Specialty cut flowers are in demand, especially in the domestic market as they can be grown with low production costs without the need for specially equipped greenhouses and offer diversity in terms of form, texture, and colour. These products, which are widely cultivated in the USA, are not well known in Türkiye. One of the main problems of the Turkish cut flower sector is its dependence on foreign inputs and the lack of product diversity. Therefore, specialty cut flower production can be an alternative crop for Türkiye, which has climatic advantages. The cut flower potential of plants such as *Zinnia elegans, Tagates erecta, Helianthus annuus, Gomphrena globosa, Centaurea cyanus*, and *Cleome spinosa* that are commonly grown in gardens has been evaluated. After harvesting these flowers grown in an open field in June-October 2020, the stem length (cm), stem thickness (mm), flower length (cm), flower diameter (cm), flower weight (g), and vase life (days) were measured. As a result of the evaluations, considering the phenological findings such as flowering and harvesting period, *Zinnia elegans, Tagates erecta, Helianthus annuus, Gomphrena globosa*, and *Centaurea cyanus* were found to be suitable for Eskişehir climatic conditions due to their long flowering periods. On the other hand, if the stem length value, which is one of the most important parameters for cut flowers, is taken as a reference, the minimum stem length value of 30 cm and above is met by *Zinnia elegans, Tagates erecta, Helianthus annuus*, and *Cleome spinosa* while the vase life value of 6 days and above is met by *Zinnia elegans, Tagates erecta, Helianthus annuus, Gomphrena globosa*, and *Cleome spinosa*. However, *Cleome spinosa* was not found to be suitable for the region due to its low yield value and short flowering period, while *Zinnia elegans, Helianthus annuus, Tagates erecta*, and *Gomphrena globosa* were found to be plants that could be evaluated for the region. In addition, it is believed that the cultivation of specialty cut flowers, with the selection of suitable species, will be an alternative production in regions without climatic advantages.

# INTRODUCTION

Cut flowers are commonly cultivated in various growing media, such as soil or soilless and solid or liquid, within greenhouses equipped with different technologies. Setting up and

providing these environments for production is very costly, especially in the start-up phase. The selection of plants that are easy to sow, plant, maintain, and harvest and that can be grown in the open field in environments with suitable ecological conditions reduces costs. Therefore, unlike traditional cut flowers (TFC) such as roses, carnations, and chrysanthemums, which require special environments and cultural treatments during cultivation specialty cut flowers (SCF), which are easy to grow and can be grown in the open field, are preferred by growers (*Cavins, 1997*). SCF cultivation requires less energy than TCF and is safer for people and the environment. In addition, the SCF group can include annual, biennial, and perennial herbaceous plants, as well as woody and bulbous plants. The wide range of products is an advantage for the sector (*Salachna, 2022*). The lower production costs of plants meeting these conditions and the greater market interest in new varieties encourage the use of SCF groups as cut flowers (*Bachmann, 2006*). On the other hand, there are gaps in the sustainability, marketing, and promotion of these groups (*Darras, 2021*). For a plant to be utilized as a cut flower, factors such as consumer demand, flowering season, growing conditions, stem length, and vase life should be considered (*Yue & Hall, 2010*; *Abdullah, Ahmad & Dole, 2020*).

SCF cultivation is widespread in the United States due to the prevalent importation of TCF from producer countries such as Ecuador and Colombia. Therefore, many local cut flower growers have opted to engage in this type of farming to remain competitive (*Yue & Hall, 2010*). In addition, the production of SCF in Australia prefers native species such as wax flower (*Chamelaucium uncinatum*), kangaroo paw (*Anigozanthos spp.*) and thryptomene (*Thryptomene spp.*), Acacia species native to Australia, such as *A. dealbata, A. retinodes*, and *A. baileyana*, which are perennial woody plants, are grown commercially in countries such as Italy and Israel, as well as Australia, because their showy flower clusters are used as cut flowers. In the USA, the market share of *Antirrhinum majus, Echinacea purpurea, Helianthus annuus, Limonium sinuatum, Matthiola incana, Scabiosa atropurpurea, Zinnia elegans*, and many other SCFs has increased (*Darras, 2021*).

According to 2022 data from TUIK, *Dianthus sp.* is the most commonly cultivated cut flower in Türkiye, followed by *Rosa sp., Chrysanthemum sp.*, and *Gerbera sp.* (*TUIK, 2023*). One of the primary issues facing Turkish cut floriculture is the restricted range of cut flower products available for sale. This situation exacerbates the country's reliance on foreign sources for production materials. Since certain species are grown in cut flower production in Türkiye, it remains behind in the world cut flower sector market despite the advantage of climatic conditions. Therefore, there is a need to increase the variety of cut flowers in local production. There are not enough studies and applications on SCF production in Türkiye. Several scientific studies have evaluated the potential for cut flowers in natural vegetation in Türkiye (*Turkmenoglu & Fakir, 2016*; *Ari & Akin, 2010*; *Yilmaz, 2012*; *Kebeli, 2021*). However, there has been no implementation of SCF production in practice. There has been a dual structure in cut flower production in Türkiye for years. One of them is the enterprise using relatively advanced technology, having large production areas, and producing for export with more professional marketing systems, while the other is the enterprise with insufficient technological equipment, producing in small areas as family businesses, and offering their products to the domestic market to a

large extent (*Ozturk, Temel & Erken, 2014*). SCFs are alternative products, especially for small family businesses producing for the domestic market. SCFs are usually grown in smaller quantities over a shorter growing season and are mostly sold at local farmers' markets as they are not as transport-resistant as TCFs and can only be stored for limited periods of time (*Landford, Curtis & Stock, 2023*).

Zinnia belongs to the Compositae family and is native to Mexico and Central America (*Riaz et al., 2008*). *Zinnia elegans*, an annual summer flower, is the best known and cultivated species of the genus Zinnia and became popular in Europe in 1790 (*Burlec et al., 2019*). *Zinnia elegans* is a short-day plant (*Boyle & Stimart, 1983*; *Park, Kim & Kim, 2013*). *Tagates erecta* is native to Central and South America, especially Mexico, and belongs to the Asteracea family (*Singh et al., 2015*). *Tagates erecta* is a commercially known species with a colour range from yellow to orange (*Aslam et al., 2016*). Helianthus is a plant native to North America that was used by Native Americans as a source of food, dye, and medicine (*Short, Etheredge & Waliczek, 2017*). Today, *Helianthus annuus* is commonly used as an ornamental plant (*Sloan & Harkness, 2006*). *Tagates erecta* and *Helianthus annuus* are facultative short-day plants (*Boyle & Stimart, 1983*). *Gomphrena glob*osa is an annual plant that belongs to the Amaranthaceae family and is native to Central America (*Ashwini, Patil & Chethan Kumar, 2019*; *Yaseen et al., 2022*). It is widely used in borders and beds. It is also used as dried flowers (*Yaseen et al., 2022*). *Gomphrena globosa* is a neutral day plant (*Boyle & Stimart, 1983*). The genus Centaurea belongs to the family Asteracea and contains about 400–500 species. Most are native to the eastern Mediterranean and some to Europe (*Kadman-Zahavi & Yahel, 1985*). *Centaurea cyanus*, an annual meadow plant with blue-violet flowers, is also known as an ornamental plant (*Yang & Zhang, 2022*; *Kulchenko et al., 2020*). *Centaurea cyanus* is characterised as a long-day plant (*Seidlová, 1964*). *Cleome spinosa* is an annual or biennial herbaceous plant of the Capparidaceae family, native to tropical America and now widely cultivated throughout the world (*Qin et al., 2012*). The species, which is widely used for medicinal purposes, is also used as an ornamental plant (*Albarello et al., 2006*). *Cleome spinosa* is a facultative short-day plant (*Currey, Lopez & Mattson, 2011*).

In this study, it was aimed at increasing the diversity in the market by evaluating the potential of some seasonal flowers to be used as cut flowers and, at the same time, popularizing this cultivation, which is possible to grow on open land and requires less cost. This study is derived from Ilkay Coskun's master thesis.

## MATERIALS AND METHODS

### Plant material and experimental design

The study was carried out between May and November 2020 in the greenhouse and fields of Eskisehir Osmangazi University, Faculty of Agriculture, whose geographical coordinates are 39°45′30″N latitude and 30°28′15″E longitude. Eskisehir has a unique climate due to the fact that it is within the influence area of Central Anatolia, the Western Black Sea, and Mediterranean climates. From mid-December to mid-February, very cold days and frosts are experienced. There are more frosts in March. In the second half of spring, the maximum temperature rises above 20 °C. In July and August, the Mediterranean shows

the characteristics of a summer drought. Rain in October and sleet in November indicate the onset of winter (*Eskisehir Provincial Directorate of Agriculture and Forestry, 2023*). *Zinnia elegans*, *Tagates erecta*, *Helianthus annuus*, *Gomphrena globosa*, *Centaurea cyanus*, and *Cleome spinosa* were used as plant material in the study. The study from seed sowing to the seedling formation stage, was carried out in the greenhouse. The next part of the study was continued in the field. The seeds were sown in the first week of May in peat-filled seedling trays and maintained in greenhouse conditions at a temperature of 20–25 °C and 60–70% humidity. Seedlings with two to four true leaves formed after four to 6 weeks were planted in the field in June. Seedlings were planted on the bed in four rows, with a spacing of 25 cm × 25 cm between and above the rows. The planting plan was established in three replications according to the randomized complete block. In each replicate, 72 plants for *Zinnia elegans*, 52 plants for *Tagates erecta*, 24 plants for *Helianthus annuus*, 44 plants for *Gomphrena globosa*, 36 plants for *Centaurea cyanus*, and 56 plants for *Cleome spinosa* were used. During planting, 1 g of 15:15:15 NPK fertilizer was applied to each planting hole. Plants were watered by drip irrigation as needed. In addition, plants were fertilised three times after planting with liquid fertiliser containing 25% humic and fulvic acid, 3% total nitrogen, 2% organic nitrogen, 4% water-soluble $K_2O$, and 40% total organic matter. During the experiment, cultural applications such as irrigation, fertilisation, and weed control were applied equally to all plants.

The stage when the petals are fully opened was chosen to harvest the flowers. Flower harvesting was carried out by cutting about 1–2 cm above the point where branching started.

## Data collection

Stem length (cm), stem thickness (mm), flower length (cm), flower diameter (mm), and fresh flower weight (g) of the flowers harvested at the full flowering stage were measured. The number of flowers harvested was recorded, and the yield per plant was calculated. In addition, the vase life of the flowers was determined. The vase life trial was carried out in three replicates, using 3–7 flowers in each replicate. Vase life was determined by keeping them in vases filled with tap water under room-temperature conditions of 22–25 °C. The stage at which approximately half of the petals have wilted is considered the end of vase life.

## Statistical analysis

The parameters examined were determined using the General Linear Model ANOVA procedure in the Minitab-7 package program. Differences between factors were revealed by the Tukey (HSD) multiple comparison test at the 1% and 5% significance levels (*Duzgunes et al., 1987*).

## Soil

The analysis of the experimental soil revealed its composition to be loamy throughout the profile. The levels of organic matter were 1.98%, pH 7.69, $P_2O_5$ 6.44 (kg/da) and $K_2O$ were 207.9 (kg/da).

**Table 1  Climatic conditions of research area.**

|  | June | July | August | September | October |
|---|---|---|---|---|---|
| Max.Temperature (°C) | 31.6 | 35.9 | 37.0 | 38.4 | 34.2 |
| Min.Temperature (°C) | 5.7 | 12.3 | 9.9 | 8.2 | 3.6 |
| Average temperature (°C) | 18.5 | 23.0 | 22.8 | 20.9 | 15.2 |
| Mean relative humidity (%) | 77.2 | 63.6 | 57.4 | 65.0 | 77.8 |
| Total insolation time (hour) | 225.9 | 364.8 | 142.3 | 248.3 | 191.3 |
| Total precipitation (mm = kg ÷ m²) | 115.8 | 0.2 | 1.0 | 3.4 | 29.0 |
| Average 20 cm soil temperature (°C) | 20.5 | 25.1 | 26.0 | 24.3 | 22.1 |

## Climate data

The climatic data of Eskisehir province between June and November 2020, including the planting and harvesting periods of the seedlings, are shown in Table 1.

# RESULTS

## Phenological results

Seeds of all plants were sown in the first week of May under greenhouse conditions. The plants, which completed seedling development in 3 to 4 weeks, were planted in the open field in the second and third weeks of June. All plants flowered in July and were harvested in the same month. The flower harvest of *Cleome spinosa* continued until the third week of August and that of *Centaurea cyanus* until mid-September. *Zinnia elegans, Tagates erecta, Helianthus annuus*, and *Gomphrena globosa* were harvested until the last weeks of October (Table 2).

## Morphological results

Statistical evaluations of the morphological measurement data of the plants are given in Table 3.

In *Zinnia elegans*, stem thickness and flower weight values were found to be statistically significant according to harvest time. Stem length varied according to the harvest time, and the highest value of 52.9 cm was obtained in the July and October flower harvests. The highest average stem length was measured in October (48.3 cm), and the lowest stem length was measured in September (47.7 cm). The highest flower length was 56.2 cm. The highest flower length was determined in October (51.4 cm), and the lowest flower length was determined in September (50.6 cm). The highest stem thickness was observed in July (4.4 mm), and the highest monthly average value (3.7 mm) was also in July. The lowest stem thickness of *Zinnia elegans* was determined in August and September (2.4–2.3 mm, respectively). Considering the monthly average, the highest value was in September with 2.7 mm. Regarding the flower diameter, the highest value of 6.7 cm was observed in the first harvesting process in July, while the lowest value was recorded again in July (4.8 cm). The highest average flower diameter was measured in September (6.2 cm),

Table 2 Phenological observation of plant materials.

| | May | June | | | | | July | | | | August | | | | | September | | | | | October | | | |
|---|---|---|---|---|---|---|---|---|---|---|---|---|---|---|---|---|---|---|---|---|---|---|---|---|
| Z.e | • | θ | θ | θ | + | ⊥ | ▲ | ❀ | ❀ | ❀ | ❀ | ❀ | ❀ | ❀ | ❀ | ❀ | ❀ | ❀ | ❀ | ❀ | ❀ | ❀ | ❀ | ❀ |
| T.e | • | θ | θ | θ | + | ⊥ | ⊥ | ▲ | ▲ | ❀ | ❀ | ❀ | ❀ | ❀ | ❀ | ❀ | ❀ | ❀ | ❀ | ❀ | ❀ | ❀ | | |
| H.a | • | θ | θ | θ | + | ⊥ | ⊥ | ⊥ | ▲ | ▲ | ❀ | ❀ | ❀ | ❀ | ❀ | ❀ | ❀ | ❀ | ❀ | ❀ | ❀ | | | |
| G.g | • | θ | θ | θ | θ | + | ⊥ | ⊥ | ▲ | ❀ | ❀ | ❀ | ❀ | ❀ | ❀ | ❀ | ❀ | ❀ | ❀ | ❀ | ❀ | | | |
| C.c | • | θ | θ | θ | + | ⊥ | ▲ | ❀ | ❀ | ❀ | ❀ | ❀ | ❀ | ❀ | ❀ | ❀ | ❀ | ❀ | | | | | | |
| C.s | • | θ | θ | θ | + | ⊥ | ▲ | ▲ | ❀ | ❀ | ❀ | ❀ | ❀ | ❀ | | | | | | | | | | |

Description
- • Sowing seeds
- θ Germination
- + Planting seedlings in the field
- ⊥ Seedling growth
- ▲ Budding
- ❀ Flowering and harvest

and the lowest average flower diameter was measured in August (5.6 cm). In the study, the fresh flower weights of *Zinnia elegans* flowers harvested at the full bloom stage were measured immediately after harvest, and results ranging from 43.7 to 17.9 g were obtained. The highest monthly average was 33.2 g in July. The flower yield per plant of *Zinnia elegans* was four flowers in the study.

Stem length, flower length, stem thickness, flower diameter, and flower weight values were found to be statistically significant according to harvest time in *Tagates erecta*. The highest stem length value was obtained in August (55.1 cm). The lowest stem length value (21.4 cm) was recorded in October. The lowest monthly average value in terms of stem length is in October. In *Tagates erecta*, the highest flower length was recorded in August (58.9 cm), and the lowest flower length (25.2 cm) was recorded in October. The highest average flower length was 50.2 cm in July, and the lowest average flower length was 33.7 cm in October. The highest stem thickness was recorded at 5.8 mm at the first harvest in July, and the lowest stem thickness was recorded at 2.3 mm in September. The highest average stem thickness was observed in July (4.6 mm), and the lowest average stem thickness was observed in October (3.2 mm). The highest flower diameter of *Tagates erecta* was 6.06 cm in August, and the lowest flower diameter was 3.63 cm in August. The highest average flower diameter was determined in October (5.15 cm), and the lowest average flower diameter was determined in August (4.64 cm). The highest fresh flower weight was measured in July (82.8 g) and the lowest in October (14.2 g). The highest average flower weight was observed in July (64.7 g), and the lowest average flower weight was observed in October (19.6 g). Flower yield per plant is 4 pieces in *Tagates erecta*.

The stem thickness value of *Helianthus annuus* was found to be statistically significant according to harvest time. The mean stem length was highest in August (44.6 cm) and lowest in October (32.2 cm). The maximum flower length was recorded in August (55.4 cm). The average flower length in August was 48.2 cm. Concerning stem thickness,

**Table 3 Morphological features of plant materials.**

| Plant | Time | Stem length (cm) | Stem thickness (mm)** | Flower length (cm) | Flower diameter (cm) | Flower weight (g)* |
|---|---|---|---|---|---|---|
| Zinnia elegans | Jul. | 48.1a | 3.7a | 50.9a | 5.7a | 33.2a |
| | Aug. | 47.9a | 3.0bc | 51.1a | 5.6a | 27.8b |
| | Sept. | 47.7a | 2.7c | 50.6a | 6.2a | 23.6ab |
| | Oct. | 48.3a | 3.6ab | 51.4a | 6.1a | 25.1ab |
| | | ns | $P < 0.01$ | ns | ns | $P < 0.05$ |
| **Plant** | **Time** | Stem length (cm)** | Stem thickness (mm)* | Flower length (cm)** | Flower diameter (cm)* | Flower weight (g)** |
| Tagates erecta | Jul. | 47.6a | 4.6a | 50.2a | 4.9ab | 64.7a |
| | Aug. | 43.0a | 3.6ab | 46.8a | 4.6ab | 40.9b |
| | Sept. | 35.5b | 3.3b | 39.3b | 4.9ab | 27.1c |
| | Oct. | 29.8b | 3.2b | 33.7b | 5.1a | 19.6c |
| | | $P < 0.01$ | $P < 0.05$ | $P < 0.01$ | $P < 0.05$ | $P < 0.01$ |
| **Plant** | **Time** | Stem length (cm) | Stem thickness (mm)* | Flower length (cm) | Flower diameter (cm) | Flower weight (g) |
| Helianthus annuus | Jul. | 44.3a | 7.1a | 46.8a | 11.2a | 81.2a |
| | Aug. | 44.6a | 6.3ab | 48.2a | 12.3a | 79.3a |
| | Sept. | 43.4a | 4.5c | 46.9a | 11.4a | 41.1a |
| | Oct. | 32.2a | 3.5bc | 35.0a | 9.3a | 27.8a |
| | | ns | $P < 0.05$ | ns | ns | ns |
| **Plant** | **Time** | Stem length (cm)** | Stem thickness (mm)* | Flower length (cm)** | Flower diameter (cm)* | Flower weight (g)** |
| Gomphrena globosa | Jul. | 36.5a | 2.3a | 38.4a | 1.8a | 16.4a |
| | Aug. | 30.0b | 1.7b | 31.9b | 1.6ab | 6.5b |
| | Sept. | 25.8b | 1.6b | 27.5b | 1.5b | 4.5b |
| | Oct. | 16.2c | 1.3b | 17.8c | 1.4b | 1.8b |
| | | $P < 0.01$ | $P < 0.05$ | $P < 0.01$ | $P < 0.05$ | $P < 0.01$ |
| **Plant** | **Time** | Stem length (cm)* | Stem thickness (mm)* | Flower length (cm)* | Flower diameter (cm)* | Flower weight (g)* |
| Centaurea cyanus | Jul. | 26.4a | 1.6a | 28.3a | 2.7a | 13.2a |
| | Aug. | 22.6ab | 1.1b | 24.9ab | 2.4b | 7.5ab |
| | Sept. | 18.9b | 0.8b | 21.0b | 2.3a | 1.3b |
| | | $P < 0.05$ | $P < 0.05$ | $P < 0.05$ | $P < 0.05$ | $P < 0.05$ |
| **Plant** | **Time** | Stem length (cm)** | Stem thickness (mm)* | Flower length (cm)** | Flower diameter (cm) | Flower weight (g)** |
| Cleome spinosa | Jul. | 40.8a | 3.6a | 51.3a | 8.3a | 16.2a |
| | Aug. | 30.3b | 2.4b | 37.3b | 8.0a | 4.3b |
| | | $P < 0.01$ | $P < 0.05$ | $P < 0.01$ | ns | $P < 0.01$ |

**Notes:**
Means with different letters within each column are significant at 0.01 and 0.05 level (**$P < 0.01$–*$P < 0.05$).
Means with same letters are not significant (ns).

the average thickness was 7.1 mm in July and 3.5 mm in October. The highest value for the flower diameter was recorded in August (16.3 cm) and the monthly average was 12.3 cm. The highest average weight of fresh flowers was recorded in July (81.2 g) and the lowest in October (18.6 g). The number of flowers per plant was 11 when the *Helianthus annuus* plant was evaluated for flower yield.

**Table 4 Vase life of plant materials.**

| | Vase life of flower (day) | | | | | |
|---|---|---|---|---|---|---|
| Time | *Zinnia elegans*** | *Tagates erecta* | *Helianthus annuus* | *Gomphrena globosa*** | *Centaurea cyanus*** | *Cleome spinosa* |
| Jul. | 10.9a | 12.8a | 9.5a | 7.5b | 5.0b | 7.0a |
| Aug. | 8.5b | 11.2a | 7.9a | 8.5ab | 5.8ab | 7.3a |
| Sept. | 12.2a | 12.0a | 7.6a | 10.8a | 6.8a | – |
| Oct. | 9.0ab | – | 7.6a | – | – | – |
| | $P < 0.01$ | ns | ns | $P < 0.05$ | $P < 0.05$ | ns |

Note:
Means with different letters within each column are significant at 0.01 and 0.05 level (**$P < 0.01$–*$P < 0.05$). Means with same letters are not significant (ns).

The stem length, stem thickness, flower length, flower diameter, and flower weight values of *Gomphrena globosa* were found to be statistically significant according to harvest time. The highest stem length was measured in July (38.7 cm) and the lowest in October (12.7 cm). According to the monthly averages, the highest value (36.5 cm) belongs to July. According to the months, the highest value (38.4 cm) of average flower length was recorded in July, while a decrease in average flower length was observed towards the end of the growing season. The highest stem thickness value was determined in July (3.0 mm), and the lowest stem thickness value was determined in September (0.9 mm). The highest average stem thickness was 2.3 mm in July and the lowest was 1.3 mm in October. Flower diameter and weight were highest in July at 1.8 mm and 16.4 g, respectively, according to the monthly averages. The flower yield of *Gomphrena globosa* is about seven flowers per plant.

Stem length, flower length, stem thickness, flower diameter, and flower weight values were found to be statistically significant according to harvest time in *Centaurea cyanus*. The stem length values ranged between 15 and 35.6 cm. The highest stem length was determined in July, while the lowest value was determined in September. According to the monthly averages, the highest value of stem length (26.4 cm) is again in July. The highest value of average stem thickness was 1.6 mm in July and 0.8 mm in September. The highest flower diameter was determined in July (3.1 cm), and the lowest flower diameter was determined in August (1.8 cm). The highest average flower diameter was obtained from the plants harvested in July (2.7 cm), while the lowest was obtained from the plants harvested in September (2.3 cm). The highest fresh flower weight was 22.7 g in July. According to the harvested months, the highest average flower weight was recorded in July (13.2 g), while flower weight decreased towards September (avg. 1.3 g). When the plant is evaluated in terms of flower yield, there are four flowers per plant.

The values for stem length, stem thickness, flower length, and flower weight of *Cleome spinosa* were found to be statistically significant according to harvest time. The maximum stem length was 46.4 cm in July and the minimum was 24 cm in August. The July average for stem length is 40.8 cm. The highest value was measured in July (5 mm) and the lowest in August (1.8 mm) when considering stem thickness. The highest average stem thickness was recorded in July (3.6 mm). The lowest average stem thickness was recorded in August (2.4 mm). The average value of the diameter of the flower according to the months is

8.3 cm in July and 8 cm in August. Monthly averages of fresh flower weight were 16.2 g in July and 4.3 g in August. The flower yield of the plant is one per plant.

## Vase life results

One of the most determinative criteria for the use of a plant as a cut flower is its vase life. The findings of the vase life values of the flowers grown within the trial are shown in Table 4. The vase lives of *Zinnia elegans, Gomphrena globosa*, and *Centuarea cyanus* flowers were statistically significant according to the harvest time. The highest vase life values for *Zinnia elegans* were observed for flowers harvested in July and September. The highest vase life values for *Gomphrena globosa* and *Centaurea cyanus* were recorded for flowers harvested in September (10.8 and 6.8 days, respectively). The vase lives of *Tagates erecta, Helianthus annuus, and Cleome spinosa* flowers were found to be statistically insignificant according to harvest time.

## DISCUSSION

The suitability of a flower for the cut flower trade depends on some parameters. A straight and strong stem, flower size, vase life, maturity, uniformity, and leaf quality are among the factors that should be used in cut flower classification (*Reid, 2004*). In addition, post-harvest resistance, easy handling, and, of course, yield are other factors besides visual factors. *Varu & Barad (2010)* stated that flower size, harvest stage, odor and freshness determine the quality of cut flowers. In the global cut flower trade, it can be said that, in general, optimal stem length and flower size, together with the maximum number of flowers are the determining criteria for a good cut flower (*Mladenović et al., 2020*). The long flower stem is a reason for preference, as it is easy to harvest. Cut flower industry standards for preferred stem length vary by species and location. For example, the minimum stem length for roses to be exported is 40 cm, while the minimum stem length for roses to be offered on the domestic market is 25 cm. Moreover, it is worth noting that the majority of cut flower distributors favour a stem length of no less than 46 cm, irrespective of the species (*Loyola, Dole & Dunning, 2019*). Cut flower growers consider the stem length to be marketable when it is between 30 and 41 cm at the minimum (*Kelly, 1991*). In our study, the minimum stem length (38.3 cm) of *Zinnia elegans* was recorded in July, the first harvest month. However, the average stem length in July was 48.1 cm. The minimum acceptable stem length for post-harvest evaluation of *Zinnia elegans* plants is considered to be 30 cm (*Martins et al., 2021*). In *Tagates erecta*, the minimum stem length (24.9 cm) was determined in October at the last harvest. The lowest average stem length (29.8 cm) according to months was also in October. *Helianthus annuus* and *Gomphrena globosa* had the lowest average stem lengths of 32.2 and 16.2 cm respectively, in October. The minimum average stem length was found in September (18.9 cm), the last harvest time for *Centaurea cyanus*, and in August (30.3 cm), the last harvest time for *Cleome spinosa*. From these data, it can be concluded that the minimum stem length values of *Zinnia elegans, Tagates erecta, Helianthus annuus*, and *Cleome spinosa* are more suitable for cut flower use, while the stem lengths of *Gomphrena globosa* decreased in September and October, but stem length averages in July and August were more suitable for cut

flowers. In *Centaurea cyanus*, the maximum length of the stem (31 cm) was measured in July. The average stem length for July was 26.4 cm. Some short-stemmed cut flowers can be used in bridal bouquets or in multiple arrangements. This creates a contrast between the different sizes.

On the other hand, *Reid (2004)* notes that objective standards such as stem length, which is still the main quality standard for many flowers, may have little relevance to flower quality, vase life, or usefulness.

When selecting species for specialty cut flower production, other important characteristics besides stem length are stem strength, early flowering, disease and pest resistance, heat tolerance, and vase life (*Kelly, 1991*). Stem thickness is usually a good indicator of the strength of the stem. A long stem with a thin stem diameter is more likely to bend or break than a shorter stem of the same thickness or a stem of the same length with a larger stem diameter (*Ortiz, Hyrczyk & Lopez, 2012*). Considering the stem thickness of the plants examined in the study, the maximum stem thickness value of *Zinnia elegans* was 4.4 mm in July, and the highest value (3.7 mm) in terms of mean stem thickness was found again in July. *Riaz et al. (2011)* reported that the stem thickness of *Zinnia* plants grown in different environments was 5.3 and 6.8 mm, while *Sardoei, Fahraji & Ghasemi (2014)* mentioned that the stem thickness values of *Zinnia elegans* plants grown in different environments were recorded between 2.2 and 3.1 mm. *Demirel et al. (2021)* stated that the stem thickness of *Zinnia elegans* plants varied between 5.5–7.3 mm at different irrigation levels. In comparison with other studies, it can be said that the value for the thickness of the stem is low. *Halagi et al. (2023)* studied the effect of planting distance and pinching on *Tagates erecta* and found the thickest stem to be 1.64 cm and the thinnest to be 1.48 cm. *Meena et al. (2015)* obtained a maximum stem thickness of 1.40 cm and a minimum stem thickness of 1.30 cm in *Tagates erecta* in their study, where they tested planting time, planting distance, and pinching treatments. *Priyadarshini, Palai & Nath (2018)* examined the effect of nitrogen applications on the development of *Tagates erecta* and reported that while the stem thickness was 0.89 cm in the control group plants, the maximum stem thickness was 1.37 cm. In the study, the maximum stem thickness of *Tagates erecta* was 5.1 mm, the maximum average was 4.6 mm in July, and the minimum was 3.2 mm in October. Compared to other studies, the stem thickness value is thought to be thin. There is no scientific study on the stem thickness of *Centaurea cyanus*.

In the study, the maximum stem thickness value of *Gomphrena globosa* was 3 mm in July, while the highest monthly average was 2.3 mm in the same month. *Green et al. (2010)* found the average stem thickness to be 2.1 mm in May and 2.7 mm in August. While the mean value for July was close to that mentioned in the previous sentence, low values for stem thickness were recorded in the other months. The desired stem thickness of *Helianthus annuus* for general floral arrangements should be 0.5–1.5 cm, according to *Sloan & Harkness (2006)*. In this study, the months of July (7.1 mm) and August (6.3 mm) met this criterion with regard to the values of the stem thickness of *Helianthus annuus*. However, it remained below this standard in other months as air temperatures began to fall. The stem thickness of *Cleome spinosa* has not been scientifically studied.

The flower diameter may vary between species and varieties, as it depends on genetic factors. The diameter of the flower is not the most important criterion for determining the quality of a cut flower. As relatively large and colourful flowers attract more attention, they can be used in bouquets in quantities of one or fewer. On the other hand, minimal flowers can be used more intensively to create bouquets or as a complement to other flowers. In this context, among the plants we used in our study, with *Helianthus annuus, Cleome spinosa, Zinnia elegans*, and *Tagates erecta*, which have the largest flower diameter, respectively, fewer plants can be used when forming bouquets. *Centaurea cyanus* and *Gomphrena globosa* can be used as filler material with their minimal flowers, or bouquets can be created using a larger number of flowers.

Vase life is a critical factor in determining the market value of cut flowers. However, since it takes several days for cut flowers to be sold from farmers to general consumers through retailers, the quality of cut flowers may decrease during this period (*Horibe, 2020*). The harvest period of the flower is also an important factor in vase life. The optimal harvest time for different species of cut flowers is different. To ensure the longest possible vase life, flowers should be harvested at the proper stage of development (*Senapati et al., 2016*).

*Iqbal et al. (2012)* examined the changes in vase life by applying different growth regulators to *Zinnia elegans* plants after harvest. They recorded the shortest vase life (6.3 days) in the control group without any treatment. The longest vase life was obtained from salicylic acid treatment, with 11.3 days. *Ahmad et al. (2012)* reported that the average vase life of *Zinnia elegans* is 13.4 days. In our study, the results of the vase life of the *Zinnia elegans* (maximum of 13.3 days, maximum monthly average 12.2 days) are satisfactory in comparison with other studies. The average maximum vase life of *Tagates erecta* was 12.8 days, while the minimum was 11.2 days. According to *Ahmad & Dole (2014)*, in their study, where they applied homemade and commercial flower preservatives to *Tagates erecta* flowers after harvest, they determined the longest vase life to be 15.7 and 15.5 days after different treatments. The lowest value was found at 7.9 days in tap water. *Halagi et al. (2023)*, the longest vase life was found to be 7.4 days in *Tagates erecta* with different planting distances and pinching treatments. In this case, the vase life results obtained from flowers kept in tap water without any treatment in our study are satisfactory.

*Hanks & Mason (2017)* stated that the average vase life of *Helianthus annuus* is between 7 and 11 days; similarly, *Kilic et al. (2020)* stated that it is 8.9 days. The results of our study (7.6–9.5 days) are consistent with these studies.

*Gomphrena globosa* flowers have a vase life of 7.5–10.8 days in our study. *Grant (2021)* states that *Gomphrena globosa* has an average vase life of 7 days, and *Saghajit (1994)* says 11 days. The values obtained in our study were found to be compatible with other studies.

Although there are not many studies on the vase life of *Centaurea cyanus*, *Sellam et al. (2016)* reported that the vase life of *Centaurea moschata* varied between 4.6 days in the control group, 6 days in sucrose treatment, and 9 and 9.6 days in plants in which various bioregulators were used. In our study, the maximum average for *Centaurea cyanus* was 6.8 days.

*Clark et al. (2010)* reported that the average vase life of *Cleome hassleriana* was 5.9 days. The average maximum vase life of *Cleome spinosa* flowers was 7.3 days in our study.

In terms of vase life values, similar results were obtained with the study of *Clark et al. (2010)*.

*Nguyen & Lim (2021)* state that each type of cut flower has a different vase life and divide the vase life into three groups: short (less than 5 days), medium (6 ~ 14 days), and long (2 ~ 4 weeks). In this respect, it can be said that all flower species evaluated in the study have medium vase life values, except for the flowers of *Centaurea cyanus* (5 days) harvested in July.

The ranking in terms of flower yield values per plant during the production season is as follows: *Helianthus annuus* (11 total flowers per plant), *Gomphrena globosa* (seven total flowers per plant), *Zinnia elegans* (four total flowers per plant), *Centaurea cyanus* (four total flowers per plant), *Tagates erecta* (four total flowers per plant), and *Cleome spinosa* (one total flower per plant). In our study, no cultural treatments such as pinching and different planting spacing were applied to increase yield. However, it has been reported that widening the planting spacing (*Armitage, 1987*) and pinching increase flower yield (*Wien, 2016*; *Ehsanullah et al., 2021*) and delay flowering (*Cheema, 2018*) in some annual ornamentals. For example, *Wien (2016)* reported that widening planting spacing and pinching positively affected yield in *Helianthus annuus*; *Ullah et al. (2019)* reported that especially double pinching improved many parameters related to yield and plant quality in *Zinnia elegans*; *Singh, Singh & Ahirwar (2018)* reported that wide plant spacing (60 cm × 60 cm) increased flower yield per plant in *Tagates erecta*. Similarly, *Nathan et al. (2019)* reported that pinching 15 days after planting and foliar application of bioregulator increased flowering and improved flower quality in *Gomphrena globosa*, while *Selahvarzi et al. (2023)* reported that pinching in *Helianthus annuus* prolonged the flowering period by about 1 week.

Phenologically, the flowering period is longer in *Zinnia elegans* (16 weeks) and *Tagates erecta* (14 weeks), followed by *Gomphrena globosa* (12 weeks), *Helianthus annuus* and *Centaurea cyanus* (11 weeks), and *Cleome spinosa* (6 weeks). The first flowering was observed in *Zinnia elegans* and *Centaurea cyanus* in the first week of June, followed by *Cleome spinosa*, *Tagates erecta*, and *Gomphrena globosa* in the same month, and *Helianthus annuus* at the end of June. Flowering is influenced by many factors, including the intrinsic characteristics of the plant genotype, the growing medium, and growing conditions (*Proietti et al., 2022*). In order to regulate flowering in open field cultivation, some practices, such as including many species in the programme during the growing season, planting in rotation, different planting times to bring flowering earlier or later, pinching, and chilling, can be done. It may also be necessary to create low tunnels to extend the growing season.

Grown in the open, these plants require no additional inputs other than fertiliser and weed control during the pre-planting and growing phases, making them more economical than traditional cut flower production (*Bachmann, 2006*). As they are easy to harvest, no special equipment is required. The fact that it can easily be done on suitable land with the supply of seeds and/or seedlings makes specialty cut flower production economical. As they are sold directly from the farm and/or florists, post-harvest handling such as transport and storage are also economical (*Landford, Curtis & Stock, 2023*). Cut flower

cultivation in Türkiye is mainly concentrated in the Aegean and Mediterranean regions due to the climatic advantage of having a Mediterranean climate type (*Baris & Uslu, 2009*; *Kenanoğlu, 2023*). However, the fact that this cultivation can be carried out in the open field, enabling cultivation in the spring and summer months, makes this cultivation possible in other regions of the country. These flowers are more economical than other imported products, especially as they are offered to the domestic market. They also increase product diversity. Considering these aspects, the economic contribution of specialty cut flower cultivation cannot be ignored.

## CONCLUSIONS

Due to climate change and the environmental, energy, and economic problems it causes, the ornamental horticulture sector, like all other sectors, needs a transformation that adopts an ecological approach. In this context, the cultivation of specialty cut flowers, which can be easily grown outdoors without the need for additional equipment, will be more ecological and economical, especially by supporting the cultivation of natural flowers. In the local market, florists are able to deliver faster and fresher flowers directly from the grower, and offer more vibrant and healthier flowers to the market. It can also offer different alternatives to consumers who are bored with classic-cut flowers and want to see different and more economical flowers. On the other hand, it is possible to carry out the production by selecting the plants that are suitable for the ecology outside the regions with a favourable climate.

The study was carried out in a region with climatic disadvantages in terms of ornamental plant cultivation and where ornamental plant cultivation is not common. The plants have been grown in the open air during the summer period. When we look at both the phenological and morphological characteristics of the flowers, it is possible to say that the development of *Zinnia elegans*, *Helianthus annuus*, *Tagates erecta*, *Gomphrena globosa*, and *Centaurea cyanus* plants is good in the Eskisehir region from spring to mid-autumn. In terms of flowering period, it was concluded that the harvest period of all flowers except the *Cleome spinosa* flower was long. In our study, it was thought that these plants, which are cultivated in order to increase the diversity of production and support local product diversity, should be brought to the cut flower market. In addition, cultivation can be supported in regions where ornamental plants are not cultivated at a low cost. It is very important that this cultivation be carried out with the right plant selection in geographies with climatic disadvantages. Therefore, the plants we evaluated in this study can be a guide for those who want to grow cut flowers in ecologies similar to Eskisehir. To extend the production season, plastic tunnels, which are more economical than greenhouses, can be used as an alternative to open-field production.

Besides, there is a need for more research on the introduction of specialty cut flowers to the ornamental plants sector throughout the country, popularisation of their cultivation and flower quality, flower yield, vase life, storage, and transport of flowers.

## Funding

This research was supported by Eskisehir Osmangazi University Scientific Research Projects Coordination Unit, A1 Master's Thesis Project (No.2020-23A1-09). The funders had no role in study design, data collection and analysis, decision to publish, or preparation of the manuscript.

## Grant Disclosures

The following grant information was disclosed by the authors:
by Eskisehir Osmangazi University Scientific Research Projects Coordination Unit, A1 Master's Thesis Project: 2020-23A1-09.

## Competing Interests

The authors declare that they have no competing interests. Altinbilek Agricultural Products Licensed Warehousing Inc. has no contribution to this study. Ilkay Coskun started to work for the company after the completion of this study.

## Author Contributions

- Sibel Yiğiter conceived and designed the experiments, performed the experiments, analyzed the data, prepared figures and/or tables, authored or reviewed drafts of the article, contributed to the statistical analysis, experiment planning, implementation and conversion into articles, and approved the final draft.
- İlkay Coskun performed the experiments, analyzed the data, prepared figures and/or tables, authored or reviewed drafts of the article, contributed to experiment application, execution and writing of the article, and approved the final draft.

## Data Availability

The raw measurements are available in the Supplemental Files.

## Supplemental Information

Supplemental information for this article can be found online at http://dx.doi.org/10.7717/peerj.17114#supplemental-information.

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
