# Peer review of "Evaluation of some garden flowers as specialty cut flowers in Eskisehir province-Türkiye"

_PeerJ, doi:10.7717/peerj.17114_

## Round 0.1 · original submission · Major Revisions

Dear Authors,

I hope this letter finds you well.

I am writing to inform you that we have received the reviewers' comments on your manuscript.

The reviewers have provided valuable feedback and suggestions to enhance your research's clarity, quality, and impact. Please carefully review the reviewers' comments and suggestions and revise your manuscript accordingly.

Please address each comment raised by the reviewers in a detailed response letter explaining the changes made.

Please carefully proofread your revised manuscript to eliminate errors or inconsistencies before submission.

Best wishes,

**Language Note:** The Academic Editor has identified that the English language must be improved. PeerJ can provide language editing services - please contact us at copyediting@peerj.com for pricing (be sure to provide your manuscript number and title). Alternatively, you should make your own arrangements to improve the language quality and provide details in your response letter. – PeerJ Staff

Reviewer 1 ·

Basic reporting

Thank you for considering me to review the manuscript titled “Evaluation of Some Garden Flowers as Specialty Cut Flowers in Eskisehir Province, Turkey.” The authors explored the potential of using certain garden flowers as cut flowers. The manuscript contributes valuable insights that could enhance the diversity of products in Turkey's cut flower industry, promoting economically and environmentally sustainable production. However, the manuscript requires substantial revisions.
Suggestions:
English language editing is necessary throughout the manuscript to rectify grammatical errors and shorten lengthy sentences.
Rewrite the abstract to be more descriptive, emphasizing the experiment's objectives, and provide additional methodological details.
Simplify the confusing sentence at lines 17-19.
Consider using "garden flowers" instead of "specialty cut flowers" to accurately represent the flowers evaluated for specialty cut flower production.
Remove “For this purpose” from line 23, as it seems unnecessary.
Avoid using the term “good,” which is vague and lacks scientific precision; instead, tailor the statement to serve the manuscript's purpose.
Specific Sections for Improvement:

Introduction:
Clarify the global scenario of specialty-cut flower production and highlight whether the types of flowers used exist in other countries but not in Turkey, or if this evaluation is novel for both Turkey and the world.
Add information about the plant species used, and elucidate the knowledge gap, rationale, and hypothesis more thoroughly.

Material and Method:
More details about the experimental site are needed including longitude, latitude, and meteorological data.
Elaborate on applied agricultural practices, and provide information about the lighting period at Line 77.
Correct “coincidence blocks” to “randomized complete block design” at Line 79.

Results:

Present soil physical and chemical properties and climatic data earlier in the Materials and Methods section, as these details don't belong in the Results.
Describe the content of Table 1 in the text without including the table itself. For example “The analysis of experimental soil revealed its composition to be loamy throughout the profile. The levels of organic matter, pH, P2O5, and K2O, were .....:
Expand upon and improve the information in Table 3 (lines 102-106) for clarity.
Ensure consistency in naming the flower species throughout the manuscript.
Revise the statistical analysis in Table 4 for accuracy. The stem lengths of Helianthus annuus at 44.3 cm and 32.2 cm should display a significant difference

Discussion:
Include recent literature in the discussion, and clarify ambiguous statements, like “good stem length” at Line 212, with precise scientific expressions.
Explain the characteristics of selected plants and discuss their potential for bouquet arrangement or other purposes besides cut flowers.
Add information about how utilizing these plants as cut flowers can contribute to the economy.

Conclusion:
Summarize and condense the conclusion while offering more general advice applicable on a global scale. Ensure consistency in formatting scientific names by italicizing "Tagetes erecta" and others throughout the manuscript.

References:
It's vital to follow the specific journal's style guide or citation requirements for uniformity and accuracy. Review and standardize the reference section according to journal style guidelines. Ensuring consistency in journal abbreviations. Some journals are abbreviated as “Acta Hortic” in line 413 while others are not abbreviated. Consistency in the inclusion of DOIs (Digital Object Identifiers) within references is essential. If the citation style you're following recommends the inclusion of DOIs for certain types of sources or journals, ensure that all relevant references contain DOIs. However, not all sources might have DOIs available. In such cases, follow the citation style guide's instructions regarding DOIs and include them only if they are available for the sources cited.
Tables:
Adjust Tables 2, 3, 4, and 5 by removing unnecessary formatting and using periods instead of commas as decimal separators.

Experimental design

Appropriate

Validity of the findings

The authors explored the potential of using certain garden flowers as cut flowers. The manuscript contributes valuable insights that could enhance the diversity of products in Turkey's cut flower industry, promoting economically and environmentally sustainable production. However, the manuscript requires substantial revisions.

·

Basic reporting

It is thought that the quality of the study will increase by making the necessary corrections. It is clear that this study will contribute to the literature.

Experimental design

no comment

Validity of the findings

no comment

Additional comments

Basic Reporting
It is thought that the quality of the study will increase by making the necessary corrections. It is clear that this study will contribute to the literature.

Abstact
Inline 28-29:
In this section, it can be more clearly stated in the sentence as a result of.. which species has cut flower potential and in what respect.
Introduction
In line 54-57:
Once the explanation of abbreviations is given in the text, there is no need to express them again, such as (SCF).
Inline 66:
Is the literature on these species sufficient, not only in Turkey but also worldwide? If it is not enough, its importance not only for Turkey but also for world literature should be emphasized. For example, has any study been done before to evaluate these species as SCF? If it has not been done, this indicates the richness that the study adds to the literature.
Results
Inline 96-98:
Are soil properties a finding? If it is a finding, it should be included in the method section; if not, it should be included in the material section.
In line 102-106:
Information regarding the examination of phenological findings should also be included in the summary section. These findings are important in studies that take into account the harvest times of the relevant species.
The Latin names of the species should be stated at the first stage when the species names are used.
Conclusion
In line 305:
In this section, its importance can be emphasized especially for geographies similar to Türkiye or Eskişehir ecology. Or the fact that these species (if any) are resistant to extreme conditions may be presented as an advantage.

·

Basic reporting

no comment

Experimental design

Some necessary additions and revisions are listed below (in Additional comments).

Validity of the findings

Some necessary additions and revisions are listed below (in Additional comments).

Additional comments

- Names of the cultivars can be mentioned.
-
- Detailed information about the planting method should be mentioned in the material method. Information such as seedlings were planted whether in single-row or bed
-
- “The study was carried out between May-November 2020 in the greenhouse and fields of Eskisehir” (It is seen that the study was conducted only under greenhouse conditions. However, this expression (in the greenhouse and fields) also means that it is carried out in greenhouse and also in open fields.
-
- “of these field-grown flowers were measured” (The expression gives the impression that the experiment was conducted outdoors rather than in the greenhouse.)
-
- instead of this sentence: “the harvesting period of all the flowers except the flower of Cleome was long” “the harvest period of all the species except Cleome, was long” may be used.
-
- “15:15:15 NPK fertiliser was applied to the planting holes.”(The amount (weight) of the fertilizer per hole should be mentioned)
-
- “During the experiment, cultural applications such as irrigation, fertilisation and weed control were applied equally to all plants”. (if an additional fertilizer were applied during the experiment (in addition to 15:15:15 NPK fertilizer applied at the begining), it should be mentioned)
-
- “Vase life was determined by keeping them in vases filled with tap water under room temperature conditions of 22-25 0C.” (It should be stated which criteria was used as for deciding the ending of vase life (eg. wilting of one or two petal).
-
- “trays and maintained in greenhouse conditions at a temperature of 20-25 oC and 60-70% humidity” (Was the greenhouse climate controlled during cultivation? Were those temperature values provided only during the seedling stage?)
-
- In the material method, at which stage were the flowers harvested (for example, at the stage when the petals flattened) and how was the flowers harvested (eg. stem was cutted from the point where the branching started from top to bottom) shoud be mentioned.
-
- “The longest vase life was determined as 17 days in September. The shortest vase life was recorded as 4” (The results have already been evaluated statistically and average values are given. So it is not necessary to give this data (17 days, 4 days) Similar statements should be corrected elsewhere)
-
- The highest average vase life of Zinnia elegans flowers according to months (12.2 days) was in September. (This statement would not be correct, because statistically July September are in the same group, so it would be more accurate to say that the highest vase life values were obtained in July and September. Similar statements should be corrected elsewhere)
-
- Yield values are stated in the results. These values seem low (e.g. 4 flowers for tagates). Are these yield values monthly or total yield? Statistics of yield data, which is an important parameter, should also be given in the table.
-
- More reference should be added to relevant literature in discussion.
-
- In the study, the results and discussions sections focused mainly on the effect of months on different types of parameters. It would be better to include statistical comparisons of at least some parameters (such as yield, vase life, earliness) between species.

---

## Round 0.2 · accepted · Accept

Dear Authors,

I am pleased to inform you that after the last round of revision, the manuscript has been improved a lot, and it can be accepted for publication.

Congratulations on accepting your manuscript, and thank you for your interest in submitting your work to PeerJ.

With Thanks

Reviewer 1 ·

Basic reporting

The authors have satisfactorily addressed my previous concerns. I recommend accepting the manuscript for publication.

Experimental design

The authors have satisfactorily addressed my previous concerns.

Validity of the findings

The authors have satisfactorily addressed my previous concerns.

·

Basic reporting

The proposed corrections have been implemented. The article is now suitable for publication.

Experimental design

The proposed corrections have been implemented. The article is now suitable for publication.

Validity of the findings

The proposed corrections have been implemented. The article is now suitable for publication.

Additional comments

The proposed corrections have been implemented. The article is now suitable for publication.

·

Basic reporting

No comment

Experimental design

No comment

Validity of the findings

No comment

Additional comments

Suggestion: In material method the explanation (in the lines 142-144) mentioned below should be given in further detail (the amount of the fertiliser should be mentioned)
"In addition, plants were fertilised three times after planting with liquid fertiliser containing 25% humic and fulvic acid, 3% total nitrogen, 2% organic nitrogen, 4% water-soluble K2O, and 40% total organic matter."